# Thromboembolic Episodes in Patients with Systemic Lupus Erythematosus Without Atrial Fibrillation/Atrial Flutter Are Related to the Presence of at Least 3 Points in the CHA_2_DS_2_-VA Score: A Comprehensive Retrospective Analysis of 787 Patients

**DOI:** 10.3390/jcm14113920

**Published:** 2025-06-03

**Authors:** Radosław Dziedzic, Michał Węgiel, Andżelika Siwiec-Koźlik, Magdalena Spałkowska, Lech Zaręba, Stanisława Bazan-Socha, Mariusz Korkosz, Joanna Kosałka-Węgiel

**Affiliations:** 1Jagiellonian University Medical College, Doctoral School of Medical and Health Sciences, św. Łazarza 16, 31-530 Kraków, Poland; radoslaw.dziedzic@doctoral.uj.edu.pl; 2University Hospital in Kraków, 2nd Department of Cardiology, Jakubowskiego 2, 30-688 Kraków, Poland; michal.jan.wegiel@gmail.com; 3University Hospital in Kraków, Department of Rheumatology, Immunology and Internal Medicine, Jakubowskiego 2, 30-688 Kraków, Poland; lek.andzelika.siwiec@gmail.com (A.S.-K.); stanislawa.bazan-socha@uj.edu.pl (S.B.-S.); mariusz.korkosz@uj.edu.pl (M.K.); 4Jagiellonian University Medical College, Department of Dermatology, Botaniczna 3, 31-501 Kraków, Poland; magdalena.spalkowska@uj.edu.pl; 5University of Rzeszów, Institute of Computer Science, Pigonia 1, 35-310 Rzeszów, Poland; lzareba@ur.edu.pl; 6Jagiellonian University Medical College, Department of Internal Medicine, Faculty of Medicine, Jakubowskiego 2, 30-688 Kraków, Poland; 7Jagiellonian University Medical College, Department of Rheumatology and Immunology, Jakubowskiego 2, 30-688 Kraków, Poland

**Keywords:** systemic lupus erythematosus, ischemic stroke, CHA_2_DS_2_-VA score, cardiovascular risk factors

## Abstract

**Background/Objectives:** Systemic lupus erythematosus (SLE) is an autoimmune disease associated with an increased prevalence of cardiac and cerebrovascular events. Despite advancements in management, no validated tools exist that can predict the risk of ischemic stroke in SLE patients. However, several studies have demonstrated an association between a higher CHA_2_DS_2_-VASc score and an enhanced risk of ischemic stroke in autoimmune diseases without atrial fibrillation (AF) or atrial flutter (AFL). Recently, the European Society of Cardiology suggested the use of a revised score of CHA_2_DS_2_-VASc without taking sex into account (CHA_2_DS_2_-VA). Therefore, we sought to check if the new CHA_2_DS_2_-VA score might predict stroke or other cardiovascular events in SLE patients without AF/AFL. **Patients and Methods**: We retrospectively analyzed the records of patients with SLE treated at the University Hospital in Kraków, Poland, from 2012 to 2022. Patients with a history of AF/AFL were excluded. **Results**: This study enrolled 787 SLE patients without AF/AFL (aged 49 (38–60) years) with a predominance of women (*n* = 705, 89.58%). Common comorbidities included arterial hypertension (*n* = 376, 47.78%) and hypercholesterolemia (*n* = 345, 43.84%). Most non-AF/AFL SLE patients had 0–1 points in the CHA_2_DS_2_-VA score (*n* = 514, 65.31%). Overall, ischemic stroke occurred in 47 cases during a median follow-up of 8 (4–17) years regarding time from the SLE diagnosis to the stroke, with the incidence rising from 0% (*n* = 0/297) to 28% (*n* = 14/50) as the CHA_2_DS_2_-VA score increased from 0 to ≥5 points. No ischemic strokes or other thromboembolic events occurred among the 575 (73.06%) patients with a CHA_2_DS_2_-VA score of 0–2 points. In the whole cohort, patients with ≥3 points in the CHA_2_DS_2_-VA score (*n* = 212, 26.94%) were older at the last visit, had longer disease duration, were more commonly of the male sex, and were diagnosed more frequently with ischemic stroke or other thromboembolic events in their medical history (*p* < 0.05, for all) compared to those with 0–2 points (*n* = 575, 73.06%). However, in multivariable logistic regression, among the CHA_2_DS_2_-VA components, only older age (≥50 years) was related to the increased risk of thromboembolic complications (OR = 2.09, 95% CI: 1.36–3.22). Other determining factors included the presence of lupus anticoagulant (OR = 3.39, 95% CI: 2.20–5.27) and neurological SLE symptoms (OR = 2.19, 95% CI: 1.19–4.02). Interestingly, male sex (OR = 0.34, 95% CI: 0.22–0.52) and general SLE symptoms (OR = 0.43, 95% CI: 0.28–0.67) were associated with a decreased risk of thromboembolic events in this model (*p* = 0.034, for the model). **Conclusions**: SLE-related factors seem important for the onset of thromboembolic episodes. However, a higher CHA_2_DS_2_-VA score may also help to identify SLE patients with an increased risk of cardiovascular events, including stroke. Prospective studies with a long-term analysis need to be validated using the CHA_2_DS_2_-VA score to predict stroke risk in SLE patients.

## 1. Introduction

Systemic lupus erythematosus (SLE) is a heterogeneous autoimmune disease that affects various organs. Common symptoms include general manifestations such as fatigue and fever, followed by arthralgia, arthritis, cutaneous and mucosal lesions, and photosensitivity. In addition to those hallmark signs, SLE can impact vital organs, such as the kidneys, heart, lungs, and nervous system [1]. While its etiology remains multifactorial, involving genetic predisposition, hormonal factors, and environmental triggers [2], the underlying pathology primarily depends on immune system dysregulation, leading to widespread inflammation and tissue damage [3,4]. Moreover, a persistent systemic inflammatory response may affect the course of concomitant diseases, including accelerated atherosclerosis, leading to increased morbidity and mortality [5]. Particular relevance has been attributed to adverse cardiac and cerebrovascular events, contributing to a worse prognosis [6]. Such complications might be driven not only by the inflammatory response, hypercoagulable state, endothelial dysfunction, and use of immunosuppressive agents, common in SLE patients, but also by classical atherosclerosis risk factors and genetic susceptibility [7]. This is evidenced by even a four-fold mortality increase in SLE patients compared to the general population [8]. Therefore, there is a need for a validated tool to predict adverse events, particularly related to cardiovascular complications [9].

The contribution of traditional cardiovascular risk factors compared to chronic inflammation related to autoimmune disease is currently difficult to quantify, given the scarcity of data [10,11,12]. Among non-traditional, SLE-specific risk factors of cardiovascular diseases, the most important are disease activity, organ damage (including lupus nephritis and neurolupus), glucocorticosteroid use, and antiphospholipid antibodies [13,14]. These factors, combined with the hypercoagulable state, endothelial dysfunction, and oxidative stress, contribute to a heightened risk of adverse cardiovascular events in SLE patients [15,16]. The CHA_2_DS_2_-VASc score, originally developed and validated to assess the likelihood of cerebrovascular events in patients with non-valvular atrial fibrillation (AF) [17,18], is also recognized as a predictor of cardiovascular events and increased mortality rate across different populations, regardless of cardiac rhythm [19,20]. It comprises various clinical parameters indicative of or affecting atherosclerosis, including heart failure, age, sex, history of arterial hypertension, diabetes mellitus, vascular disease, and ischemic stroke. However, the European Society of Cardiology (ESC) recently recommended shifting from the CHA_2_DS_2_-VASc to the CHA_2_DS_2_-VA score, excluding female sex as an independent risk factor [21]. Female sex is now considered an age-dependent stroke risk modifier rather than a standalone risk factor. This has prompted a move toward a more inclusive and simplified risk assessment for healthcare professionals.

Several analyses have demonstrated an association between higher CHA_2_DS_2_-VASc score and an increased risk of ischemic stroke in patients with autoimmune diseases [5,6]. Still, data on the SLE population remain scarce, with no outcomes regarding the new CHA_2_DS_2_-VA score and stroke risk in patients suffering from autoimmune diseases. Therefore, we sought to evaluate the utility of the CHA_2_DS_2_-VA score in predicting ischemic stroke among SLE patients without AF/AFL and investigate how different scores can characterize SLE patients.

## 2. Patients and Methods

### 2.1. Study Population

We conducted a retrospective review of the medical records for all SLE cases diagnosed and treated at the University Hospital in Kraków, Poland, from January 2012 to June 2022. At the time of the data collection, all patients met the current European Alliance of Associations for Rheumatology/American College of Rheumatology (EULAR/ACR) 2019 classification criteria [22].

We retrieved from the medical records data on demographic and clinical characteristics, including sex, current age, age at first SLE symptoms, and diagnosis, duration of symptoms before diagnosis, disease duration, clinical and laboratory manifestations of SLE, comorbidities, medication, and cause and age of death (if applicable). The recorded treatment modalities included glucocorticosteroids, hydroxychloroquine or chloroquine, azathioprine, methotrexate, cyclosporine, mycophenolate mofetil, cyclophosphamide, immunoglobulins intravenously in suppressive doses, and biological agents (belimumab, rituximab, and anifrolumab) or plasmapheresis.

The CHA_2_DS_2_-VA score was calculated at the time of SLE diagnosis by adding 1 point each for the presence of congestive heart failure, history of arterial hypertension, diabetes mellitus, vascular disease, and age ≥ 65 years and 2 points each for age ≥ 75 years and history of ischemic stroke or transient ischemic attack, as stated in the recently published ESC recommendations [21]. Congestive heart failure was defined as a left ventricular ejection fraction below 40%, as determined by echocardiography, accompanied by signs and symptoms indicative of right or left ventricle failure [23]. Arterial hypertension was determined by a history of blood pressure ≥ 140/90 mmHg or current antihypertensive treatment. Diabetes mellitus was defined as fasting serum glucose above 7.0 mmol/L or the use of insulin or hypoglycemic agents. Vascular disease encompassed prior myocardial infarction, peripheral artery disease, or the presence of aortic plaque. The diagnosis of ischemic stroke, including TIA, is based on the assessment of clinical symptoms during the transient episode and the findings from neuroimaging studies [24,25]. In turn, SLE patients with AF/AFL in medical history were excluded from this study.

The Bioethics Committee of the Jagiellonian University Medical College has approved the research (No: 118.6120.41.2023, on 15 June 2023). All procedures adhered to the ethical principles outlined in the Declaration of Helsinki.

### 2.2. Laboratory Analysis

Anti-nuclear antibodies (ANA) were evaluated by an indirect immunofluorescence (IIF) technique using HEp-2 cells. Anti-Sjögren’s-syndrome-related antigen A (SSA), anti-Sjögren’s-syndrome-related antigen B (SSB), anti-histone, anti-nucleosome, anti-Smith (Sm), and anti-ribonucleoprotein (RNP) autoantibodies were identified by an enzyme-linked immunosorbent assay (ELISA) or a line-blot immunoassay. Anti-double-stranded DNA (anti-dsDNA) antibodies were assayed by IIF using *Crithidia luciliae* as a substrate. Serum complement levels (C3c and C4) and rheumatoid factor (RF) were assessed by nephelometry. Laboratory tests for hypercoagulability included lupus anticoagulant (LA), anticardiolipin (aCL), and anti-beta-2-glycoprotein I (anti-β2GPI) antibodies, both in IgM and IgG classes. Briefly, diluted Russell’s viper venom time (dRVVT; LA1-screen; Siemens, Germany) and a sensitive aPTT (PTT LA; Diagnostica Stago, France) were used for screening, whereas LA2-confirm (Siemens, Germany) and Staclot LA (Diagnostica Stago, USA) were run for the confirmation. Reference values for each test were established using the 99th percentile of the healthy population. Commercially available immunoenzymatic assays were applied to determine anticardiolipin (aCL) and anti-β2-glycoprotein I (anti-β2GPI) antibodies (QUANTA Lite^®^ aCL and anti-β2GPI (Inova Diagnostics, San Diego, CA, USA)).

### 2.3. Statistical Analysis

We used STATISTICA Tibco 13.3 software (StatSoft Inc., Tulsa, OK, USA) and R program (version 4.3.1.) to analyze the data. Categorical variables were presented as numbers with percentages and compared using the Chi^2^ test or Fisher’s exact test, as appropriate. The Shapiro-Wilk test was used to assess the normality of the data distribution. All continuous variables were non-normally distributed and thus presented as medians with Q1–Q3 ranges and compared with the Mann–Whitney test. For the binary variables, the odds analysis (with a 95% confidence interval [CI]) was applied to calculate the odds ratio (OR). To develop a model distinguishing group 1 (patients with any thromboembolic episode) from group 0 (patients without thromboembolic episodes), we built a multiple logistic regression model using the Akaike criterion for the best fit. A significance threshold of two-sided *p*-values below 0.05 was used for all results.

## 3. Results

### 3.1. Patient Characteristics

A summary of the demographic parameters of the SLE cohort is provided in Table 1. This study included 787 SLE patients without AF/AFL. Among the study participants, 652 (82.85%), 108 (13.72%), and 27 (3.43%) were aged ≤65 years, 65–74 years, and ≥75 years, respectively. The median age at the last visit was 49 (38–60) years. As expected, women constituted most cases in the study (*n* = 705, 89.58%). The most common comorbidities were arterial hypertension (*n* = 376, 47.78%), followed by hypercholesterolemia (*n* = 345, 43.84%) and vascular disease (*n* = 199, 25.29%). The most frequently utilized treatments were glucocorticosteroids (*n* = 734, 93.27%), chloroquine or hydroxychloroquine (*n* = 608, 77.26%), and azathioprine (*n* = 291, 36.98%).

Most non-AF/AFL SLE patients had 0–1 points (*n* = 514, 65.31%) in the CHA_2_DS_2_-VA score. However, about one-fourth (*n* = 212, 26.94%) had at least three points (Table 2).

### 3.2. Detailed Characteristics of the Subgroup with Ischemic Stroke

We did not observe ischemic strokes in non-AF/AFL SLE patients with 0–2 points in the CHA_2_DS_2_-VA score (*n* = 575, 73.06%). In turn, ischemic strokes were reported in 47 cases of those with at least three points (*n* = 212, 26.94%) during the median follow-up from SLE diagnosis of 8 (4–17) years. The incidence of the first ischemic stroke increased from 0% (*n* = 0/297) to 28% (*n* = 14/50) as the CHA_2_DS_2_-VA score increased from 0 to ≥5 points (Table 3).

First, we characterized patients with ischemic stroke (*n* = 47) compared to those without any thromboembolic episodes (*n* = 616). Detailed data on this topic are presented in Table 4. The SLE subgroup with ischemic stroke was older at the last visit, had a higher CHA_2_DS_2_-VA score, and showed a higher prevalence of vascular disease and hypercholesterolemia. Notably, antiphospholipid antibodies, such as lupus anticoagulant and aCL antibodies of both classes, were also more frequently observed in this subgroup.

### 3.3. Characteristics of the Study Cohort Regarding CHA_2_DS_2_-VA Score Results

Then, based on the interesting finding that ischemic stroke occurred only in patients with at least 3 points in the CHA_2_DS_2_-VA score, we divided the whole cohort of 787 non-AF/AFL SLE patients into cases with 0–2 points and those with at least 3 points in the CHA_2_DS_2_-VA score for further analysis (Table 5). The subgroup with the higher scores (≥3 points) was characterized by longer disease duration, older age at the last visit, and more frequent representation of the male sex compared to those with lower scores (0–2 points). As expected, comorbidities included in the CHA_2_DS_2_-VA score, such as congestive heart failure, arterial hypertension, diabetes mellitus, vascular disease, and cerebrovascular incidents, were also more prevalent in the subgroup with higher scores. Also, antiphospholipid antibodies occurred in this group more frequently.

### 3.4. Cerebrovascular Events, Transient Ischemic Attacks, or Other Thromboembolic Episodes in the Whole SLE Cohort

Then, we studied whether the previously reported findings would be maintained if all of the documented thromboembolic events (*n* = 171, 21.73% of the non-AF/AFL SLE patients) were analyzed (Table 6). Once again, the incidence increased from 0% (*n* = 0/297) to 98% (*n* = 49/50) as the CHA_2_DS_2_-VA score increased from 0 to ≥5 points, with no episodes in those with 0–2 points (*n* = 575, 73.06%).

Then, we performed a combined analysis of those with any thromboembolic complication in medical history, including stroke, transient ischemic attack, or other thromboembolic events (*n* = 171/787, 21.73%), as compared to the remaining (*n* = 616/787, 78.27%) (Table 7). Interestingly, the subgroup with such complications was again characterized by a higher CHA_2_DS_2_-VA score, mainly related to arterial hypertension and/or vascular disease. These patients were also older, suffered from SLE for longer, and more frequently represented the male sex. Furthermore, they more often had hypercholesterolemia. Interestingly, the ANA profile was heterogeneous between both of the analyzed subgroups, with antiphospholipid antibodies more often in those with thromboembolic complications.

Further analysis in a subgroup of 171 non-AF/AFL SLE patients with any thromboembolic events revealed that those with cerebrovascular complications, including transient ischemic attack, had similar CHA_2_DS_2_-VA scores but more frequent internal medicine comorbidities, such as hypercholesterolemia, and were more often treated with cyclophosphamide compared to those with other thromboembolic episodes (Table 8). On the other hand, the latter subgroup was diagnosed more frequently with malignant tumors.

Finally, we performed a multivariable logistic regression analysis to identify the factors associated with thromboembolic episodes in patients with SLE. We included only statistically significant determinants in this model (Table 9). Among predictors of thromboembolic events, the most important were age (≥50 years, OR = 2.09, 95% CI: 1.36–3.22, *p* < 0.001), presence of lupus anticoagulant (OR = 3.39, 95% CI: 2.20–5.27, *p* < 0.001), and neurological SLE symptoms (OR = 2.19, 95% CI: 1.19–4.02, *p* = 0.012). On the other hand, the male sex and the presence of SLE general symptoms, such as fever and weakness, were associated with a lower risk of thromboembolic complications (OR = 0.34, 95% CI: 0.22–0.52, *p* < 0.001 and OR = 0.43, 95% CI: 0.28–0.67, *p* < 0.001, respectively). However, the model demonstrated only moderate fitting, with an area under the curve of 0.72, overall accuracy of 73%, specificity of 85%, and sensitivity of 46%.

## 4. Discussion

The present study demonstrates insights into the CHA_2_DS_2_-VA score to characterize SLE patients without AF/AFL. To the best of our knowledge, this is the first study evaluating the risk factors of ischemic stroke in SLE patients without AF/AFL in a Polish cohort of SLE patients using a renewed score, as suggested by the European Society of Cardiology. These findings shed light on several key risk factors of cardiovascular events, with potential implications for developing a more personalized treatment approach.

In our study, the median (Q1–Q3 ranges) age of SLE patients at the last visit was 49 (38–60) years, and women made up 89.66% of the participants. Regarding demographic factors, our findings are consistent with previous studies, which further supports the validity and accuracy of this Polish cohort [6,26]. Interestingly, prior investigations have identified a range of predictors for adverse cardiac and cerebrovascular events among SLE patients [6,26,27,28]. These predictors can be categorized into traditional risk factors, such as male sex, cigarette smoking, hyperlipidemia, family history of cardiac disease, hypertension, malignancy, and SLE-related features, such as the presence of autoantibodies, organ damage, and SLE activity [6,26,27,28,29]. Our results partly align with those of other authors, further emphasizing that certain features of the CHA_2_DS_2_-VA score in non-AF/AFL SLE patients, such as arterial hypertension, may be associated with the development of ischemic stroke episodes in this group [6,26,27,28]. Additionally, we underlined that more frequent thromboembolic episodes and malignancies also characterized non-AF/AFL SLE patients with higher points in the CHA_2_DS_2_-VA score. However, data regarding the exact role of CHA_2_DS_2_-VA score in ischemic stroke risk stratification among non-AF/AFL SLE patients are scarce. Particularly, data are available on the previously used CHA_2_DS_2_-VASc score [6]. As presented in a paper by Wei-Syun et al. [6], most non-AF/AFL SLE patients had a CHA_2_DS_2_-VASc score equaling 1–2 points. These data are actually in line with our outcomes since we used the CHA_2_DS_2_-VA score, where no additional point is given for the female sex, and most SLE patients were women. This study also revealed that non-AF/AFL SLE patients with a CHA_2_DS_2_-VASc score ≥ 2 points vs. those with <2 points had adjusted HR = 4.06 (95% CI 3.07–5.35) and 3.85 (95% CI 2.46–6.02) during a ≤5 years and >5 years follow-up regarding ischemic strokes, respectively [6]. Notably, no ischemic strokes or other thromboembolic events were observed among the 575 patients with a CHA_2_DS_2_-VA score of 0–2. That novel and clinically significant finding suggests that the risk of such events can be associated with a higher CHA_2_DS_2_-VA score. However, the multivariable logistic regression model also identified SLE-related factors as strongly associated with thromboembolic episodes, such as the presence of lupus anticoagulant and neurological SLE involvement.

Predictably, patients who had experienced an ischemic stroke also showed the presence of antiphospholipid antibodies, consistent with the existing literature [30]. Furthermore, we found a higher prevalence of anticardiolipin antibodies (IgG/IgM) and lupus anticoagulant in patients with a higher CHA_2_DS_2_-VA score (≥3 points) and in those with thromboembolic episodes, which were related to each other.

Finally, we used the CHA_2_DS_2_-VA score as recommended by the recently published ESC guidelines [21]. However, this scale was prepared to guide decision-making on initiating oral anticoagulation (OAC) therapy in AF patients, advised for individuals with a CHA_2_DS_2_-VA score of ≥2, and considered with a score of 1, independently of sex. Thus, it was not validated for other groups of patients, and our results must be taken cautiously.

We would like to acknowledge some of the limitations of our research. The retrospective nature of this study may introduce inherent biases in data collection and patient selection, potentially impacting the accuracy of the findings. Additionally, the single-center design may limit the generalizability of the results to a broader population. Detailed personal lifestyle behaviors, such as smoking status, alcohol use, physical activity, and dietary patterns, were not included in this database. Moreover, crucial laboratory data, including inflammatory markers, markers of SLE activity at the time of ischemic stroke onset, and even the precise time of the thromboembolic events, were unavailable in this cohort. That made reliable time-to-event analysis, such as Cox regression model or Kaplan–Meier curves, impossible. Finally, some observed relationships, particularly in subgroup comparisons, may be incidental and not represent a cause-and-effect relationship.

## 5. Conclusions

SLE-related factors seem to be essential for the onset of thromboembolic episodes in this disease. However, a higher CHA_2_DS_2_-VA score may also be helpful to identify SLE patients with an increased risk of cardiovascular events, including stroke. The findings underscore the need for a personalized and thorough analysis of the risk factors, but further prospective and mechanistic studies are crucial to understanding the development of ischemic stroke or other thromboembolic events in SLE patients.

## Figures and Tables

**Table 1 jcm-14-03920-t001:** Baseline characteristics of analyzed in the study systemic lupus erythematosus patients without atrial fibrillation/atrial flutter.

Characteristics	SLE Patients Without AF/AFL *n* = 787
**Basic disease characteristics**	
Age at SLE diagnosis, years	33 (24–46)
Age at last visit, years	49 (38–60)
Follow-up, years from SLE diagnosis	8 (4–17)
**Frequency of age of SLE patients at last visit**	
Age at last visit ≤ 65 years, *n* (%)	675 (85.77%)
Age at last visit 65–74 years, *n* (%)	88 (11.18%)
Age at last visit ≥ 75 years, *n* (%)	24 (3.05%)
**Sex of SLE patients**	
Female, *n* (%)	705 (89.58%)
Male, *n* (%)	82 (10.42%)
**Underlying disease (components of the CHA_2_DS_2_-VA)**	
Congestive heart failure, *n* (%)	23 (2.92%)
Arterial hypertension, *n* (%)	376 (47.78%)
Diabetes mellitus, *n* (%)	65 (8.26%)
Cerebrovascular accident, or transient ischemic attack, or another thromboembolic episode, *n* (%)	171 (21.73%)
Vascular disease ^1^, *n* (%)	199 (25.29%)
**CHA_2_DS_2_-VA**	
CHA_2_DS_2_-VA score, points	1 (0–3)
**Other underlying disease**	
Hypothyroidism, *n* (%)	191 (24.27%)
Hyperthyroidism, *n* (%)	38 (4.82%)
Hypercholesterolemia ^2^, *n* (%)	345 (43.84%)
End-stage kidney disease, *n* (%)	10 (1.27%)
MGUS, *n* (%)	7 (0.89%)
Malignant tumor, *n* (%)	63 (8.01%)
**Immunosuppressive treatment**	
Glucocorticoids oral and/or intravenous, *n* (%)	734 (93.27%)
Chloroquine or hydroxychloroquine, *n* (%)	608 (77.26%)
Azathioprine, *n* (%)	291 (36.98%)
Methotrexate, *n* (%)	144 (18.30%)
Cyclosporine, *n* (%)	61 (7.75%)
Belimumab, *n* (%)	33 (4.19%)
Mycophenolate mofetil, *n* (%)	241 (30.62%)
Cyclophosphamide, *n* (%)	208 (26.43%)
Rituximab, *n* (%)	22 (2.80%)
Immunoglobulins, *n* (%)	22 (2.80%)
Plasmapheresis, *n* (%)	21 (2.67%)
Anifrolumab, *n* (%)	8 (1.02%)

Categorical variables are presented as numbers with percentages, whereas continuous variables are presented as medians with Q1–Q3 ranges. ^1^—angiographically significant coronary artery disease, history of myocardial infarction, peripheral arterial atherosclerotic disease, and atherosclerotic plaque in the aorta; ^2^—LDL > 3 mmol/L or pharmacotherapy with statin or a diagnosis based on medical history. Abbreviations: AF—atrial fibrillation, AFL—atrial flutter, SLE—systemic lupus erythematosus, *n*—number, MGUS—monoclonal gammopathy of undetermined significance, and CHA_2_DS_2_-VA score—a scoring system used to assess the risk of stroke in patients with atrial fibrillation, considering factors such as heart failure, hypertension, age, diabetes, stroke history, and vascular disease.

**Table 2 jcm-14-03920-t002:** Results of the CHA_2_DS_2_-VA score in systemic lupus erythematosus patients without atrial fibrillation/atrial flutter.

Points in the CHA_2_DS_2_-VA Score	SLE Patients Without AF/AFL *n* = 787
0 points, *n* (%)	297 (37.74%)
1 point, *n* (%)	217 (27.57%)
2 points, *n* (%)	61 (7.75%)
3 points, *n* (%)	93 (11.81%)
4 points, *n* (%)	69 (8.77%)
5 points, *n* (%)	30 (3.10%)
6 points, *n* (%)	16 (2.03%)
7 points, *n* (%)	3 (0.38%)
8 points, *n* (%)	1 (0.13%)

Categorical variables are presented as numbers with percentages. Abbreviations: AF—atrial fibrillation, AFL—atrial flutter, SLE—systemic lupus erythematosus, *n*—number, and CHA_2_DS_2_-VA score—a scoring system used to assess the risk of stroke in patients with atrial fibrillation, considering factors such as heart failure, hypertension, age, diabetes, stroke history, and vascular disease.

**Table 3 jcm-14-03920-t003:** Results of the CHA_2_DS_2_-VA score in systemic lupus erythematosus patients without atrial fibrillation/atrial flutter regarding the presence of ischemic stroke in medical history.

Points in the CHA_2_DS_2_-VA Score	SLE Patients Without AF/AFL *n* = 787	Events of Ischemic Stroke in SLE patients Without AF/AFL (Frequency Regarding All Events) *n* = 47	Frequency of Ischemic Stroke in SLE Patients Without AF/AFL (Frequency Regarding all SLE Patients with a Specific CHA_2_DS_2_-VA)
0 points, *n* (%)	297 (37.74%)	0 (0.00%)	0/297 (0.00%)
1 point, *n* (%)	217 (27.57%)	0 (0.00%)	0/217 (0.00%)
2 points, *n* (%)	61 (7.75%)	0 (0.00%)	0/61 (0.00%)
3 points, *n* (%)	93 (11.82%)	18 (38.30%)	18/93 (19.35%)
4 points, *n* (%)	69 (8.77%)	15 (31.91%)	15/69 (21.74%)
≥5 points, *n* (%)	50 (6.35%)	14 (29.79%)	14/50 (28.00%)

Categorical variables are presented as numbers with percentages. Abbreviations: AF—atrial fibrillation, AFL—atrial flutter, SLE—systemic lupus erythematosus, *n*—number, and CHA_2_DS_2_-VA score—a scoring system used to assess the risk of stroke in patients with atrial fibrillation, considering factors such as heart failure, hypertension, age, diabetes, stroke history, and vascular disease.

**Table 4 jcm-14-03920-t004:** Comparison of systemic lupus erythematosus patients without atrial fibrillation/atrial flutter with ischemic stroke as compared to those without thromboembolic episodes.

Parameter	SLE Patients Without AF/AFL Without Any Thromboembolic Episode *n* = 616	SLE Patients Without AF/AFL with Ischemic Stroke *n* = 47	*p*-Value
**Basic disease characteristics**			
Disease duration, years	12.0 (6.0–19.0)	15.5 (7.0–22.0)	0.28
Age at last visit, years	48.0 (37.0–58.0)	53.0 (47.0–63.0)	**0.004**
CHA_2_DS_2_-VA score, *n*	1 (0–1)	4 (3–5)	**<0.001**
Female sex, *n* (%)	560 (90.91%)	42 (89.36%)	0.61
**Frequency of age of SLE patients at last visit**			
Age at last visit ≤ 65 years, *n* (%)	540 (87.66%)	37 (78.72%)	0.19
Age at last visit 65–74 years, *n* (%)	57 (9.25%)	8 (17.02%)	
Age at last visit ≥ 75 years, *n* (%)	19 (3.08%)	2 (4.26%)	
**Underlying disease (components of the CHA_2_DS_2_-VA score)**			
Congestive heart failure, *n* (%)	18 (2.92%)	1 (2.13%)	1.00
Arterial hypertension, *n* (%)	275 (44.64%)	27 (57.45%)	0.10
Diabetes mellitus, *n* (%)	48 (7.79%)	6 (12.77%)	0.26
Vascular disease ^1^, *n* (%)	138 (22.40%)	19 (40.43%)	**0.005**
**Other underlying disease**			
Hypothyroidism, *n* (%)	152 (24.72%)	9 (19.50%)	0.42
Hyperthyroidism, *n* (%)	26 (4.23%)	4 (8.51%)	0.19
Hypercholesterolemia ^2^, *n* (%)	249 (40.49%)	31 (65.96%)	**0.001**
End-stage kidney disease, *n* (%)	8 (1.30%)	0 (0.00%)	0.92
MGUS, *n* (%)	4 (0.65%)	1 (2.13%)	0.27
Malignant tumor, *n* (%)	44 (7.14%)	3 (6.38%)	0.16
Lupus nephritis, *n* (%)	200 (32.73%)	15 (33.33%)	1.00
**Laboratory parameter**			
Rheumatoid factor, *n* (%)	122 (35.78%)	6 (12.77%)	**0.001**
Anti-SSA antibodies ^3^, *n* (%)	350 (60.66%)	22 (52.38%)	0.22
Anti-SSB antibodies ^3^, *n* (%)	172 (29.81%)	10 (23.81)	0.39
Anti-histone antibodies ^3^, *n* (%)	142 (24.61%)	14 (33.33%)	0.23
Anti-nucleosome antibodies ^3^, *n* (%)	174 (30.16%)	15 (35.71%)	0.39
Anti-Smith antibodies ^3^, *n* (%)	70 (12.17%)	7 (16.67%)	0.47
Anti-RNP antibodies ^3^, *n* (%)	126 (21.88%)	11 (26.19%)	0.53
Anti-dsDNA antibodies ^3^, *n* (%)	198 (34.32%)	20 (47.62%)	0.05
Anti-dsDNA antibodies ^4^, *n* (%)	358 (66.92%)	30 (73.17%)	0.40
Lupus anticoagulant, *n* (%)	76 (19.79%)	20 (46.51%)	**<0.001**
Anticardiolipin antibodies IgG or IgM, *n* (%)	218 (48.99%)	35 (77.78%)	**<0.001**
Anticardiolipin antibodies IgG, *n* (%)	144 (32.58%)	25 (55.56%)	**0.001**
Anticardiolipin antibodies IgM, *n* (%)	151 (34.32%)	24 (53.33%)	**0.009**
Anti-β2 glycoprotein I IgG or IgM, *n* (%)	69 (20.60%)	11 (27.50%)	0.31
Anti-β2 glycoprotein I IgG, *n* (%)	48 (14.59%)	6 (15.00%)	0.94
Anti-β2 glycoprotein I IgM, *n* (%)	44 (13.29%)	7 (18.00%)	0.33
**Immunosuppressive treatment**			
Glucocorticoids oral and/or intravenous, *n* (%)	576 (94.27%)	45 (95.75%)	0.93
Chloroquine or hydroxychloroquine, *n* (%)	477 (78.58%)	34 (72.34%)	0.31
Azathioprine, *n* (%)	215 (35.48%)	25 (54.35%)	**0.016**
Methotrexate, *n* (%)	109 (18.05%)	6 (13.33%)	0.39
Cyclosporine, *n* (%)	49 (8.09%)	2 (4.44%)	0.37
Belimumab, *n* (%)	30 (4.98%)	0 (0.00%)	0.12
Mycophenolate mofetil, *n* (%)	189 (31.29%)	12 (26.67%)	0.51
Cyclophosphamide, *n* (%)	159 (26.28%)	17 (36.96)	0.12
Rituximab, *n* (%)	16 (2.65%)	2 (4.44%)	0.47
Immunoglobulins, *n* (%)	16 (2.65%)	0 (0.00%)	0.26
Plasmapheresis, *n* (%)	13 (2.16%)	0 (0.00%)	0.32
Anifrolumab, *n* (%)	4 (0.66%)	0 (0.00%)	0.59

Categorical variables are presented as numbers with percentages; continuous variables are presented as medians with Q1–Q3 ranges. Statistically significant results are bolded. ^1^—angiographically significant coronary artery disease, history of myocardial infarction, peripheral arterial atherosclerotic disease, and atherosclerotic plaque in the aorta; ^2^—LDL > 3 mmol/L or pharmacotherapy with statin or a diagnosis based on medical history; ^3^—immunoblotting assay; ^4^—the *Crithidia luciliae* immunofluorescence test (CLIFT). Abbreviations: AF—atrial fibrillation, AFL—atrial flutter, anti-dsDNA—anti-double-stranded DNA, SLE—systemic lupus erythematosus, *n*—number, MGUS—monoclonal gammopathy of undetermined significance, and CHA_2_DS_2_-VA score—a scoring system used to assess the risk of stroke in patients with atrial fibrillation, considering factors such as heart failure, hypertension, age, diabetes, stroke history, and vascular disease.

**Table 5 jcm-14-03920-t005:** Characteristics of systemic lupus erythematosus patients without atrial fibrillation/atrial flutter regarding the sum of points in the CHA_2_DS_2_-VA score (0–2 points vs. ≥3 points).

Parameter	SLE Patients Without AF/AFL with a CHA_2_DS_2_-VA Score of 0–2 Points *n* = 575	SLE Patients Without AF/AFL with a CHA_2_DS_2_-VA Score of ≥3 Points *n* = 212	*p*-Value
**Basic disease characteristics**			
Disease duration, years	12.0 (7.0–19.0)	15.0 (6.0–23.5)	**0.027**
Age at last visit, years	46.0 (36.0–56.0)	57.0 (45.0–68.0)	**<0.001**
Events of ischemic stroke, *n* (%)	0 (0%)	47 (22.17%)	**<0.001**
Female sex, *n* (%)	525 (91.30%)	180 (84.91%)	**0.009**
**Frequency of age of SLE patients at last visit**			
Age at last visit ≤ 65 years, *n* (%)	532 (92.52%)	143 (67.45%)	**<0.001**
Age at last visit 65–74 years, *n* (%)	41 (7.13%)	47 (22.17%)	
Age at last visit ≥ 75 years, *n* (%)	2 (0.35%)	22 (10.38%)	
**Underlying disease (components of the CHA_2_DS_2_-VA score)**			
Congestive heart failure, *n* (%)	9 (1.57%)	14 (6.60%)	**<0.001**
Arterial hypertension, *n* (%)	235 (40.87%)	141 (66.51%)	**<0.001**
Diabetes mellitus, *n* (%)	33 (5.74%)	32 (15.09%)	**<0.001**
Any thromboembolic episode, *n* (%)	0 (0%)	171 (80.66%)	**<0.001**
Vascular disease ^1^, *n* (%)	119 (20.70%)	80 (37.74%)	**<0.001**
**Other underlying disease**			
Hypothyroidism, *n* (%)	140 (24.39%)	51 (24.71%)	0.95
Hyperthyroidism, *n* (%)	23 (4.01%)	15 (7.11%)	0.07
Hypercholesterolemia ^2^, *n* (%)	224 (39.02%)	121 (57.08%)	**<0.001**
End-stage kidney disease, *n* (%)	7 (1.22%)	3 (1.42%)	0.74
MGUS, *n* (%)	4 (0.70%)	3 (1.42%)	0.39
Malignant tumor, *n* (%)	34 (5.91%)	29 (13.68%)	**<0.001**
Lupus nephritis, *n* (%)	190 (33.33%)	63 (30.14%)	0.40
**Laboratory parameter**			
Rheumatoid factor, *n* (%)	113 (35.42%)	31 (28.18%)	0.17
Anti-SSA antibodies ^3^, *n* (%)	330 (61.11%)	105 (53.03%)	**0.048**
Anti-SSB antibodies ^3^, *n* (%)	160 (29.63%)	44 (22.22%)	**0.046**
Anti-histone antibodies ^3^, *n* (%)	137 (25.37%)	56 (28.28%)	0.43
Anti-nucleosome antibodies ^3^, *n* (%)	165 (30.56%)	73 (36.87%)	0.10
Anti-Smith antibodies ^3^, *n* (%)	64 (11.90%)	28 (14.14%)	0.41
Anti-RNP antibodies ^3^, *n* (%)	121 (22.45%)	39 (19.70%)	0.42
Anti-dsDNA antibodies ^3^, *n* (%)	188 (34.82%)	85 (42.93%)	**0.043**
Anti-dsDNA antibodies ^4^, *n* (%)	338 (67.20%)	133 (70.00%)	0.48
Lupus anticoagulant, *n* (%)	74 (20.16%)	84 (49.70%)	**<0.001**
Anticardiolipin antibodies IgG or IgM, *n* (%)	207 (48.82%)	124 (67.39%)	**<0.001**
Anti-β_2_ glycoprotein I antibodies IgG or IgM, *n* (%)	67 (20.94%)	54 (33.75%)	**0.002**
**Immunosuppressive treatment**			
Glucocorticoids oral and/or intravenous, *n* (%)	541 (94.91%)	193 (91.04%)	**0.045**
Chloroquine or hydroxychloroquine, *n* (%)	448 (79.15%)	160 (75.47%)	0.27
Azathioprine, *n* (%)	208 (36.81%)	83 (39.52%)	0.49
Methotrexate, *n* (%)	105 (18.65%)	39 (18.66%)	1.00
Cyclosporine, *n* (%)	49 (8.67%)	12 (5.74%)	0.18
Belimumab, *n* (%)	30 (5.34%)	3 (1.44%)	**0.016**
Mycophenolate mofetil, *n* (%)	185 (32.86%)	56 (26.67%)	0.10
Cyclophosphamide, *n* (%)	151 (26.77%)	57 (27.01%)	0.95
Rituximab, *n* (%)	14 (2.49%)	8 (3.83%)	0.32
Immunoglobulins, *n* (%)	16 (2.84%)	6 (2.87%)	0.98
Plasmapheresis, *n* (%)	13 (2.31%)	8 (3.83%)	0.25
Anifrolumab, *n* (%)	4 (0.71%)	4 (1.91%)	0.22

Categorical variables are presented as numbers with percentages; continuous variables are presented as medians with Q1–Q3 ranges. Statistically significant results are bolded. ^1^—angiographically significant coronary artery disease, history of myocardial infarction, peripheral arterial atherosclerotic disease, and atherosclerotic plaque in the aorta; ^2^—LDL > 3 mmol/L or pharmacotherapy with statin or a diagnosis based on medical history; ^3^—immunoblotting assay; ^4^—the *Crithidia luciliae* immunofluorescence test (CLIFT). Abbreviations: AF—atrial fibrillation, AFL—atrial flutter, anti-dsDNA—anti-double-stranded DNA, SLE—systemic lupus erythematosus, *n*—number, MGUS—monoclonal gammopathy of undetermined significance, and CHA_2_DS_2_-VA score—a scoring system used to assess the risk of stroke in patients with atrial fibrillation, considering factors such as heart failure, hypertension, age, diabetes, stroke history, and vascular disease.

**Table 6 jcm-14-03920-t006:** Detailed characteristics of systemic lupus erythematosus patients without atrial fibrillation/atrial flutter depending on the presence of any thromboembolic episodes and CHA_2_DS_2_-VA score results.

Points in the CHA_2_DS_2_-VA Score	SLE Patients Without AF/AFL *n* = 787	Events of Any Thromboembolic Episodes in SLE Patients Without AF/AFL (Frequency Regarding All Events) *n* = 171	Frequency of Any Thromboembolic Episodes in SLE Patients Without AF/AFL (Frequency Regarding All SLE Patients with a Specific CHA_2_DS_2_-VA)
0 points, *n* (%)	297 (37.74%)	0 (0.00%)	0/297 (0.00%)
1 point, *n* (%)	217 (27.57%)	0 (0.00%)	0/217 (0.00%)
2 points, *n* (%)	61 (7.75%)	0 (0.00%)	0/61 (0.00%)
3 points, *n* (%)	93 (11.82%)	63 (36.84%)	63/93 (67.74%)
4 points, *n* (%)	69 (8.77%)	59 (34.50%)	59/69 (85.51%)
≥5 points, *n* (%)	50 (6.35%)	49 (28.65%)	49/50 (98.00%)

Categorical variables are presented as numbers with percentages. Abbreviations: AF—atrial fibrillation, AFL—atrial flutter, SLE—systemic lupus erythematosus, *n*—number, and CHA_2_DS_2_-VA score—a scoring system used to assess the risk of stroke in patients with atrial fibrillation, considering factors such as heart failure, hypertension, age, diabetes, stroke history, and vascular disease.

**Table 7 jcm-14-03920-t007:** Characteristics of systemic lupus erythematosus patients without atrial fibrillation/atrial flutter regarding the presence of cerebrovascular events, transient ischemic attack, or other thromboembolic episodes versus the remaining.

Parameter	SLE Patients Without AF/AFL Without Thromboembolic Episodes *n* = 616	SLE Patients Without AF/AFL with Any Thromboembolic Episode *n* = 171	*p*-Value
**Basic disease characteristics**			
Disease duration, years	12.0 (6.0–19.0)	15.0 (6.0–23.5)	**0.008**
Age at last visit, years	48.0 (37.0–58.0)	52.0 (43.0–63.0)	**<0.001**
CHA_2_DS_2_-VA score, *n*	1 (0–1)	4 (3–5)	**<0.001**
Female sex, *n* (%)	560 (90.91%)	145 (84.80%)	**0.021**
**Frequency of age of SLE patients at last visit**			
Age at last visit ≤ 65 years, *n* (%)	540 (87.66%)	135 (78.95%)	**0.008**
Age at last visit 65–74 years, *n* (%)	57 (9.25%)	31 (18.13%)	
Age at last visit ≥ 75 years, *n* (%)	19 (3.08%)	5 (2.92%)	
**Underlying disease (components of the CHA_2_DS_2_-VA score)**			
Congestive heart failure, *n* (%)	18 (2.92%)	5 (2.92%)	1.00
Arterial hypertension, *n* (%)	275 (44.64%)	101 (59.06%)	**<0.001**
Diabetes mellitus, *n* (%)	48 (7.79%)	17 (9.94%)	0.37
Vascular disease ^1^, *n* (%)	138 (22.40%)	61 (35.67%)	**<0.001**
**Other underlying disease**			
Hypothyroidism, *n* (%)	152 (24.72%)	39 (22.94%)	0.69
Hyperthyroidism, *n* (%)	26 (4.23%)	12 (7.06%)	0.13
Hypercholesterolemia ^2^, *n* (%)	249 (40.49%)	96 (56.14%)	**<0.001**
End-stage kidney disease, *n* (%)	8 (1.30%)	2 (1.17%)	1.00
MGUS, *n* (%)	4 (0.65%)	3 (1.75%)	0.18
Malignant tumor, *n* (%)	44 (7.14%)	19 (11.11%)	0.09
Lupus nephritis, *n* (%)	200 (32.73%)	53 (31.55%)	0.77
**Laboratory parameter**			
Rheumatoid factor, *n* (%)	122 (35.78%)	22 (25.00%)	0.06
Anti-SSA antibodies ^3^, *n* (%)	350 (60.66%)	85 (52.80%)	0.07
Anti-SSB antibodies ^3^, *n* (%)	172 (29.81%)	32 (19.88%)	**0.013**
Anti-histone antibodies ^3^, *n* (%)	142 (24.61%)	51 (31.68%)	0.07
Anti-nucleosome antibodies ^3^, *n* (%)	174 (30.16%)	64 (39.75%)	**0.021**
Anti-Smith antibodies ^3^, *n* (%)	70 (12.17%)	22 (13.67%)	0.61
Anti-RNP antibodies ^3^, *n* (%)	126 (21.88%)	34 (21.12%)	0.84
Anti-dsDNA antibodies ^3^, *n* (%)	198 (34.32%)	75 (46.59%)	**0.004**
Anti-dsDNA antibodies ^4^, *n* (%)	358 (66.92%)	113 (71.52%)	0.28
Lupus anticoagulant, *n* (%)	76 (19.79%)	82 (53.95%)	**<0.001**
Anticardiolipin antibodies IgG or IgM, *n* (%)	218 (48.99%)	113 (69.33%)	**<0.001**
Anticardiolipin antibodies IgG, *n* (%)	144 (32.58%)	89 (55.98%)	**<0.001**
Anticardiolipin antibodies IgM, *n* (%)	151 (34.32%)	72 (45.57%)	**<0.001**
Anti-β2 glycoprotein I IgG or IgM, *n* (%)	69 (20.60%)	52 (35.86%)	**<0.001**
Anti-β2 glycoprotein I IgG, *n* (%)	48 (14.59%)	32 (23.19%)	**<0.001**
Anti-β2 glycoprotein I IgM, *n* (%)	44 (13.29%)	36 (26.28%)	**<0.001**
**Immunosuppressive treatment**			
Glucocorticoids oral and/or intravenous, *n* (%)	576 (94.27%)	158 (92.40%)	0.37
Chloroquine or hydroxychloroquine, *n* (%)	477 (78.58%)	131 (76.61%)	0.58
Azathioprine, *n* (%)	215 (35.48%)	76 (44.97%)	**0.024**
Methotrexate, *n* (%)	109 (18.05%)	35 (20.83%)	0.41
Cyclosporine, *n* (%)	49 (8.09%)	12 (7.14%)	0.69
Belimumab, *n* (%)	30 (4.98%)	3 (1.79%)	0.08
Mycophenolate mofetil, *n* (%)	189 (31.29%)	52 (30.77%)	0.90
Cyclophosphamide, *n* (%)	159 (26.28%)	49 (28.82%)	0.51
Rituximab, *n* (%)	16 (2.65%)	6 (3.57%)	0.53
Immunoglobulins, *n* (%)	16 (2.65%)	6 (3.57%)	0.53
Plasmapheresis, *n* (%)	13 (2.16%)	8 (4.76%)	0.07
Anifrolumab, *n* (%)	4 (0.66%)	4 (2.38%)	0.05

Categorical variables are presented as numbers with percentages; continuous variables are presented as medians with Q1–Q3 ranges. Statistically significant results are bolded. ^1^—angiographically significant coronary artery disease, history of myocardial infarction, peripheral arterial atherosclerotic disease, and atherosclerotic plaque in the aorta; ^2^—LDL > 3 mmol/L or pharmacotherapy with statin or a diagnosis based on medical history; ^3^—immunoblotting assay; ^4^—the *Crithidia luciliae* immunofluorescence test (CLIFT). Abbreviations: AF—atrial fibrillation, AFL—atrial flutter, anti-dsDNA—anti-double-stranded DNA, SLE—systemic lupus erythematosus, *n*—number, MGUS—monoclonal gammopathy of undetermined significance, and CHA_2_DS_2_-VA score—a scoring system used to assess the risk of stroke in patients with atrial fibrillation, considering factors such as heart failure, hypertension, age, diabetes, stroke history, and vascular disease.

**Table 8 jcm-14-03920-t008:** Characteristics of systemic lupus erythematosus patients without atrial fibrillation/atrial flutter regarding the presence of cerebrovascular accident or transient ischemic attack vs. other thromboembolic episodes in medical history.

Parameter	SLE Patients Without AF/AFL with Cerebrovascular Accident or Transient Ischemic Attack *n* = 65	SLE Patients Without AF/AFL with Other Thromboembolic Episodes *n* = 106	*p*-Value
**Basic disease characteristics**			
Age at SLE diagnosis, years	34.5 (26.0–50.0)	33.5 (25.0–46.5)	0.63
Disease duration, years	16.0 (7.0–26.5)	14.5 (6.0–22.0)	0.62
Age at last visit, years	53.0 (46.0–63.0)	52.0 (42.0–63.0)	0.49
CHA_2_DS_2_-VA score, *n*	4 (3–5)	4 (3–5)	0.75
Female sex, *n* (%)	58 (89.23%)	87 (82.08%)	0.21
**Frequency of age of SLE patients at last visit**			
Age at last visit ≤ 65 years, *n* (%)	52 (80.00%)	79 (74.53%)	0.72
Age at last visit 65–74 years, *n* (%)	11 (16.92%)	23 (21.70%)	
Age at last visit ≥ 75 years, *n* (%)	2 (3.08%)	4 (3.77%)	
**Underlying disease (components of the CHA_2_DS_2_-VA score)**			
Congestive heart failure, *n* (%)	2 (3.08%)	3 (2.83%)	1.00
Arterial hypertension, *n* (%)	40 (61.54%)	61 (57.55%)	0.61
Diabetes mellitus, *n* (%)	7 (10.77%)	10 (9.43%)	0.78
Vascular disease ^1^, *n* (%)	26 (40.00%)	35 (33.02%)	0.36
**Other underlying disease**			
Hypothyroidism, *n* (%)	12 (18.46%)	27 (25.71%)	0.27
Hyperthyroidism, *n* (%)	6 (9.23%)	6 (5.71%)	0.38
Hypercholesterolemia ^2^, *n* (%)	44 (67.69%)	52 (49.06%)	**0.017**
End-stage kidney disease, *n* (%)	0 (0.00%)	2 (1.89%)	0.53
MGUS, *n* (%)	1 (1.54%)	2 (1.89%)	1.00
Malignant tumor, *n* (%)	3 (4.62%)	16 (15.09%)	**0.044**
Lupus nephritis, *n* (%)	22 (34.92%)	31 (29.52%)	0.47
**Laboratory parameter**			
Rheumatoid factor, *n* (%)	9 (25.00%)	13 (25.00%)	1.00
Anti-SSA antibodies ^3^, *n* (%)	31 (51.67%)	54 (53.47%)	0.83
Anti-SSB antibodies ^3^, *n* (%)	12 (20.00%)	20 (19.80%)	0.98
Anti-histone antibodies ^3^, *n* (%)	19 (31.67%)	32 (31.68%)	1.00
Anti-nucleosome antibodies ^3^, *n* (%)	20 (33.33%)	44 (43.56%)	0.20
Anti-Smith antibodies ^3^, *n* (%)	10 (16.67%)	12 (11.88%)	0.39
Anti-RNP antibodies ^3^, *n* (%)	18 (28.33%)	17 (16.83%)	0.08
Anti-dsDNA antibodies ^3^, *n* (%)	26 (43.33%)	49 (48.52%)	0.52
Anti-dsDNA antibodies ^4^, *n* (%)	41 (71.93%)	72 (71.29%)	0.93
Lupus anticoagulant, *n* (%)	27 (45.00%)	55 (59.78%)	0.07
Anticardiolipin antibodies IgG or IgM, *n* (%)	45 (73.77%)	68 (66.67%)	0.34
Anticardiolipin antibodies IgG, *n* (%)	33 (54.10%)	56 (57.14%)	0.71
Anticardiolipin antibodies IgM, *n* (%)	29 (47.54%)	43 (44.33%)	0.69
Anti-β2 glycoprotein I IgG or IgM, *n* (%)	16 (28.57%)	36 (40.45%)	0.15
Anti-β2 glycoprotein I IgG, *n* (%)	10 (18.52%)	22 (26.19%)	0.30
Anti-β2 glycoprotein I IgM, *n* (%)	10 (18.87%)	26 (30.95%)	0.12
**Immunosuppressive treatment**			
Glucocorticoids oral and/or intravenous, *n* (%)	63 (96.92%)	95 (89.62%)	0.08
Chloroquine or hydroxychloroquine, *n* (%)	49 (75.39%)	82 (77.36%)	0.77
Azathioprine, *n* (%)	34 (53.13%)	42 (40.00%)	0.10
Methotrexate, *n* (%)	10 (15.87%)	25 (23.81%)	0.22
Cyclosporine, *n* (%)	4 (6.35%)	8 (7.62%)	0.76
Belimumab, *n* (%)	0 (0.00%)	3 (2.86%)	0.29
Mycophenolate mofetil, *n* (%)	20 (31.75%)	32 (30.19%)	0.83
Cyclophosphamide, *n* (%)	25 (39.06%)	24 (22.64%)	**0.022**
Rituximab, *n* (%)	2 (3.18%)	4 (3.81%)	0.83
Immunoglobulins, *n* (%)	1 (1.59%)	5 (4.76%)	0.41
Plasmapheresis, *n* (%)	0 (0.00%)	8 (7.62%)	**0.026**
Anifrolumab, *n* (%)	1 (1.59%)	3 (2.86%)	1.00

Categorical variables are presented as numbers with percentages; continuous variables are presented as medians with Q1–Q3 ranges. Statistically significant results are bolded. ^1^—angiographically significant coronary artery disease, history of myocardial infarction, peripheral arterial atherosclerotic disease, and atherosclerotic plaque in the aorta; ^2^—LDL > 3 mmol/L or pharmacotherapy with statin or a diagnosis based on medical history; ^3^—immunoblotting assay; ^4^—the *Crithidia luciliae* immunofluorescence test (CLIFT). Abbreviations: AF—atrial fibrillation, AFL—atrial flutter, anti-dsDNA—anti-double-stranded DNA, SLE—systemic lupus erythematosus, *n*—number, MGUS—monoclonal gammopathy of undetermined significance, and CHA_2_DS_2_-VA score—a scoring system used to assess the risk of stroke in patients with atrial fibrillation, considering factors such as heart failure, hypertension, age, diabetes, stroke history, and vascular disease.

**Table 9 jcm-14-03920-t009:** Logistic regression model on the risk of thromboembolic episodes in the analyzed cohort of patients with systemic lupus erythematosus.

Variable	β (95% CI)	OR (95% CI)	*p*-Value for the Model
Sex (0—female, 1—male)	−1.08 (from −1.52 to −0.65)	0.34 (0.22–0.52)	**0.034**
Age ≥ 50 years	0.73 (from 0.31 to 1.17)	2.09 (1.36–3.22)	
Lupus anticoagulant	1.22 (from 0.79 to 1.66)	3.39 (2.20–5.27)	
General symptoms	−0.84 (from −1.29 to −0.39)	0.43 (0.28–0.67)	
Neurological symptoms	0.78 (from 0.17 to 1.39)	2.19 (1.19–4.02)	

For the binary variables, the odds analysis (with a 95% confidence interval [CI]) was applied to calculate the odds ratio (OR). To develop a model distinguishing group 1 (patients with any thromboembolic episode) from group 0 (patients without thromboembolic episodes), we built a multiple logistic regression model using the Akaike criterion for the best fit. Statistically significant results is bolded.

## Data Availability

The data presented in this study are available on reasonable request from the corresponding author.

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
