# Peer review of "Thromboembolic Episodes in Patients with Systemic Lupus Erythematosus Without Atrial Fibrillation/Atrial Flutter Are Related to the Presence of at Least 3 Points in the CHA2DS2-VA Score: A Comprehensive Retrospective Analysis of 787 Patients"

_jcm, 2025, doi:10.3390/jcm14113920_

Round 1

Reviewer 1 Report

Comments and Suggestions for Authors

Thank you for this valuable research, although some points need to be fixed, such as:

- The objective of the study  to evaluate the CHA2DS2-VASc score's utility in predicting ischemic stroke in SLE patients without atrial fibrillation or atrial flutter. Although this score is designed for detecting cardiovascular events, do you make changes to the score?

- What is the explanation of this " with shorter median survival in patients with lower scores than those patients with higher scores" mention in line 41

- In Table 1, the age of patients at the last visit is less than the age at the time of diagnosis. Is it right?

- It is better to add raw in the table for patients receiving combined treatment

-In table 2 before % must mention the number

-In line 196 the number with stroke 36 while in table 47 which correct?

-In figure 1 what are the dots denotes?

-in line 208 number of patients have stroke and score 0-3 is 13 while in table 2 21

- In line 242, mention lower female to male although the reverse is true

-In tables and supplementary tables there are numbers like 1,2 indicate what?

-Anti-dsDNA antibodies mention two time in each table with different numbers in front of it

Comments on the Quality of English Language

The English could be improved

Author Response

Manuscript ID: jcm-3542434

Title: “Usefulness of the CHA2DS2-VA Score to Characterize Patients with Systemic Lupus Erythematosus without Atrial Fibrillation/Atrial Flutter: Insights from a Single-Center Retrospective Study”

Reviewer 1

Thank you for this valuable research, although some points need to be fixed, such as:

General response to comments:

The authors sincerely appreciate Reviewer 1 for the thorough evaluation of our work and for offering insightful suggestions that have enhanced the quality of our manuscript. Below, we have provided point-by-point responses to the comments.

- The objective of the study  to evaluate the CHA2DS2-VASc score's utility in predicting ischemic stroke in SLE patients without atrial fibrillation or atrial flutter. Although this score is designed for detecting cardiovascular events, do you make changes to the score?

Response:

Thank you very much for your comment. As suggested by another Reviewer, we revised the entire manuscript and recalculated our results using the updated CHA2DS2-VA score, as recommended in the recent guidelines by the European Society of Cardiology (ESC). Following these recommendations, the sex category was excluded (i.e., the additional point for the female sex), which is no longer considered in the updated score. We have also revised the title of the manuscript.

Our study aimed to explore the potential utility of this cardiovascular risk stratification tool in a unique patient population, that means those with systemic lupus erythematosus (SLE) but without atrial fibrillation (AF) or atrial flutter (AFL). While we acknowledge that the CHA2DS2-VA score was originally developed to guide anticoagulant therapy decision-making in patients with AF, growing evidence supports its application in broader contexts, such as predicting ischemic stroke risk in non-AF populations with chronic inflammatory diseases. Therefore, we applied the ESC-recommended version of the score in our SLE cohort to assess whether it could serve to identify patients at elevated risk of ischemic stroke, even in the absence of AF or AFL. We believe this approach provides additional insight into the cardiovascular risk profile of SLE patients and may help inform clinical decision-making in this complex group.

- What is the explanation of this " with shorter median survival in patients with lower scores than those patients with higher scores" mention in line 41

Response:

Thank you for pointing this out. The statement in this line refers to the Kaplan-Meier analysis. The interpretation is that non-AF/AFL SLE patients with lower scores (0-3 points in the CHA2DS2-VA score) developed strokes earlier than non-AF/AFL SLE patients with higher scores (> 3 points in the CHA2DS2-VA score); median time to develop strokes: 2.0 vs. 7.0 years, which is a quiet surprising outcome. We have clarified this issue in the Abstract and Results section.

- In Table 1, the age of patients at the last visit is less than the age at the time of diagnosis. Is it right?

Response:

Thank you very much for your comment, and we sincerely apologize for this mistake. The discrepancy in Table 1 occurred during the editing process while adapting the manuscript to the journal’s template. We have now corrected the data. Indeed, the age at the time of SLE diagnosis is much lower than the age at the last visit (33 [24-46] vs. 49 [38-60], years). We have also thoroughly reviewed the remaining results, and all values are now accurate and correctly presented in the revised version of the manuscript.

- It is better to add raw in the table for patients receiving combined treatment

Response:

Thank you very much for your suggestion. In our analysis, the treatment modalities for each patient were summarized and presented collectively in the table containing the general characteristics of the analyzed cohort. This approach reflects the overall therapeutic profile of the study population. Additionally, treatment details were further analyzed in subgroup comparisons.

-In table 2 before % must mention the number

Response:

Thank you very much for your comment. Table 2 has been improved accordingly.

-In line 196 the number with stroke 36 while in table 47 which correct?

Response:

Thank you very much for your comment. In total, we recorded 47 ischemic strokes in our patient cohort. However, in 11 of these cases, the exact timing of the stroke in relation to the SLE diagnosis was not available, and we were unable to retrieve the necessary detailed data. For this reason, only 36 patients were included in the further time-related survival and subgroup analyses. We have clarified that issue in the improved manuscript version.

-In figure 1 what are the dots denotes?

Response:

Thank you very much for your comment. In Figure 1, we originally presented cumulative numbers of ischemic strokes occurring at the time of SLE diagnosis or later in a group of 27 non-AF/AFL patients. The dots were initially intended to mark data points. However, to avoid unnecessary confusion, we have removed them and instead used a continuous line to more clearly illustrate the number of strokes and the timing of their occurrence. Additionally, we have included a new figure (Figure 2), which presents the cumulative number of ischemic strokes according to subgroup analysis based on CHA2DS2-VA scores.

-in line 208 number of patients have stroke and score 0-3 is 13 while in table 2 21

Response:

Thank you very much for your comment. The discrepancy results from the different patient subsets used in each analysis. In Table 2, we present the full distribution of CHA2DS2-VA scores for all 47 patients who experienced an ischemic stroke. However, in the time-related survival and subgroup analysis, only 36 patients were included, as the exact timing of the stroke relative to SLE diagnosis was not available in 11 cases. Furthermore, among these 36 patients, 9 experienced a stroke prior to their SLE diagnosis and were therefore excluded from the Kaplan-Meier curves. As a result, the final analysis included 27 patients with stroke occurring at or after SLE diagnosis. Following the recalculation using the updated version of the CHA2DS2-VA score, we identified 12 patients in the 0–3 point group and 15 patients in the >3 point group. We have clarified these distinctions in the revised Results section to ensure consistency and transparency.

- In line 242, mention lower female to male although the reverse is true

Response:

Thank you very much for your suggestion. We have revised the statement in line 242 to avoid confusion regarding the female-to-male ratio, as the data in the table are presented as n (%). In fact, the subgroup of patients with any thromboembolic complication showed a lower proportion of women compared to the group without any thromboembolic complication. We have clarified this in the Results section of the revised manuscript.

-In tables and supplementary tables there are numbers like 1,2 indicate what?

Response:

Thank you very much for your suggestion. These numbers refer to the different laboratory methods used to measure specific antibodies: (1) Immunoblotting assay and (2) the Crithidia luciliae immunofluorescence test (CLIFT). We have clarified that issue in the table footer.

-Anti-dsDNA antibodies mention two time in each table with different numbers in front of it

Response:

Thank you very much for your comment. Indeed, in our paper, anti-dsDNA antibodies were measured using two different methods: (1) Immunoblotting assay and (2) the Crithidia luciliae immunofluorescence test (CLIFT). We have clarified that issue in the table footer.

We hope that the current version of the manuscript is suitable for publication. Once again, thank you very much for your comments and suggestions.

Reviewer 2 Report

Comments and Suggestions for Authors

It is not the first time that a score validated for a certain type of patients is used in other types of patients for whom it is not validated. This is the case of CHA2DS2-VASc score, validated for patients with atrial fibrillation (AF), which has been used in non AF patients. It is well known that, while it can provide some prognostic information, its use outside AF should be interpreted with caution. Because the score was designed specifically for AF-related thromboembolism, it may not fully capture stroke risk in non-AF populations. Previous studies (also some quoted by authors) have suggested that CHAâ‚‚DSâ‚‚-VASc may predict stroke risk in other patients even without AF. However, this remains controversial. In summary, this score may overestimate or underestimate stroke risk in patients without AF and alternative risk stratification tools specific to the patient’s condition should be considered.

But even if we accept to do this intellectual exercise, authors should remind that, after years of discussion, recent ESC guidelines have deleted sex from the score. Now, the  CHAâ‚‚DSâ‚‚-VA is recommended (Eur Heart J. 2024 Sep 29;45(36):3314-3414. doi: 10.1093/eurheartj/ehae176). This is relevant because about 90% of SLE patients in this study are females. So, sex may have influenced results while female sex is no longer considered a risk factor for stroke (at least in AF).

A major problem is that CHA2DS2-VASc  changes over time and to evaluate its role authors should clearly give its values at the start of follow-up. Do follow-up start at the moment of diagnosis of SLE? This is not clearly reported in the text

Specific comments

page 1, line 41: replace higer with higher

page 2, lines 83-84: the sentence can be omitted because its content has been given a few lines above

page 2, line 89: replace characterized with characterize

page 3, line 97 : "...criteria from 2019.."??

page 3, lines 114-116: Hypercholesterolemia is not a component of CHA2DS2-VASc score so it should be reported at the end

page 4, tab 1, first 3 lines: I cannot understand ! If there is a f-up of 8 years, the age of the last visit should be much higher than the age at diagnosis

page 5, lines 179-180 : I guess that in 90% of these patients  1 point is due to sex (see general comment)

page 6, line 191: Table 3 not 2

Table 3, last column on the right: the sum of all percentages is not 100 (please check)

page 6, line 195 and page 7, line 221 Do the two paragraph  have the same title?

page 6, line 198 Better: "The other 27 strokes occurred at the time of SLE diagnosis or later"

page 6, line 199: ...after the SLE diagnosis... Delete "of"

page 11, line 282: ".. demonstrate insights of the usage .."    ??????

page 11, lines 297-298: again I repeat that hypercholesterolemia is not a feature of CHA2DS2-VASc score

page 11, line 301 : internal comorbidities ?????

page 11, lines 307-308: please specify that HR is for stroke

page 12, line 311: hazard for stroke?

page 12, lines 313-314 : sentence to be rephrased

page 12, lines 322-327: all this part should be clarified

page 12, lines 338-339: obviously I do not agree

Comments on the Quality of English Language

On average, English is quite good but style can be improved

Author Response

Manuscript ID: jcm-3542434

Title: “Usefulness of the CHA2DS2-VA Score to Characterize Patients with Systemic Lupus Erythematosus without Atrial Fibrillation/Atrial Flutter: Insights from a Single-Center Retrospective Study”

Reviewer 2

It is not the first time that a score validated for a certain type of patients is used in other types of patients for whom it is not validated. This is the case of CHA2DS2-VASc score, validated for patients with atrial fibrillation (AF), which has been used in non AF patients. It is well known that, while it can provide some prognostic information, its use outside AF should be interpreted with caution. Because the score was designed specifically for AF-related thromboembolism, it may not fully capture stroke risk in non-AF populations. Previous studies (also some quoted by authors) have suggested that CHAâ‚‚DSâ‚‚-VASc may predict stroke risk in other patients even without AF. However, this remains controversial. In summary, this score may overestimate or underestimate stroke risk in patients without AF and alternative risk stratification tools specific to the patient’s condition should be considered.

But even if we accept to do this intellectual exercise, authors should remind that, after years of discussion, recent ESC guidelines have deleted sex from the score. Now, the  CHAâ‚‚DSâ‚‚-VA is recommended (Eur Heart J. 2024 Sep 29;45(36):3314-3414. doi: 10.1093/eurheartj/ehae176). This is relevant because about 90% of SLE patients in this study are females. So, sex may have influenced results while female sex is no longer considered a risk factor for stroke (at least in AF).

A major problem is that CHA2DS2-VASc  changes over time and to evaluate its role authors should clearly give its values at the start of follow-up. Do follow-up start at the moment of diagnosis of SLE? This is not clearly reported in the text

General response to comments:

The authors sincerely appreciate Reviewer 2 for the thorough evaluation of our work and for offering insightful remarks that have enhanced our manuscript's quality.

We have recalculated the entire study using the updated CHAâ‚‚DSâ‚‚-VA score, as suggested by the Reviewer. However, our main outcomes remained unchanged, indicating the increased risk of thromboembolic events in those with higher CHAâ‚‚DSâ‚‚-VA scores. We have also revised the title of the manuscript.

The score was calculated at the time point of the SLE diagnosis. We clarified that issue in the improved manuscript version.

We agree with the Reviewer’s comment regarding the limitations of using the CHA2DS2-VA score in non-AF populations. We have clarified that issue in the revised manuscript version.

Below, we have provided point-by-point responses to the Reviewer’s 2 comments.

Specific comments

page 1, line 41: replace higer with higher

Response:

The typo has been corrected.

page 2, lines 83-84: the sentence can be omitted because its content has been given a few lines above

Response:

The manuscript has been improved, accordingly

page 2, line 89: replace characterized with characterize

Response:

The manuscript has been improved, accordingly

page 3, line 97 : "...criteria from 2019.."??

Response:

Thank you very much for your comment. We have revised the sentence to avoid any misunderstanding and clarified that the patients met the current European League Against Rheumatism and American College of Rheumatology (EULAR/ACR) 2019 classification criteria.

page 3, lines 114-116: Hypercholesterolemia is not a component of CHA2DS2-VASc score so it should be reported at the end

Response:

Thank you very much for your comment. The manuscript has been improved accordingly.

page 4, tab 1, first 3 lines: I cannot understand ! If there is a f-up of 8 years, the age of the last visit should be much higher than the age at diagnosis

Response:

Thank you very much for your comment, and we sincerely apologize for this mistake. The discrepancy in Table 1 occurred during the editing process while adapting the manuscript to the journal’s template. We have now corrected the data. Indeed, the age at the time of SLE diagnosis is much lower than the age at the last visit (33 [24-46] vs. 49 [38-60], years). We have also thoroughly reviewed the remaining results, and all values are now accurate and correctly presented in the revised version of the manuscript.

page 5, lines 179-180 : I guess that in 90% of these patients  1 point is due to sex (see general comment)

Response:

Thank you very much for your comment. We have revised the entire manuscript using the updated CHAâ‚‚DSâ‚‚-VA score, in which the female sex is no longer included as a component. After recalculating the scores, most patients had 0–1 points (n = 514, 65.31%). The updated scores are presented in Table 2.

page 6, line 191: Table 3 not 2

Response:

Thank you very much for your comment. We have corrected the table number accordingly and reviewed all tables and figures throughout the manuscript to ensure consistency.

Table 3, last column on the right: the sum of all percentages is not 100 (please check)

Response:

Thank you very much for your suggestion. We have thoroughly reviewed and corrected the table following recalculations using the updated CHAâ‚‚DSâ‚‚-VA score.

In the first column, we now present the number of points in the CHAâ‚‚DSâ‚‚-VA score (ranging from 0 to ≥ 5). The second column shows the number and percentage of SLE patients without AF/AFL with each score (percentages in this column sum to 100%). The third column provides the number of ischemic stroke events among patients with a given CHAâ‚‚DSâ‚‚-VA score, along with the percentage relative to all stroke events (percentages in this column also sum to 100%).

However, the last column presents the percentage of patients with a specific score who experienced an ischemic stroke (i.e., within-row frequency). As these percentages refer to different denominators in each row (not the total), the values in this column do not sum to 100%, which is correct by design. To clarify this, we added an explanation directly in the table. For example, for ≥ 5 points: 14 strokes among 50 patients (28.00%).

page 6, line 195 and page 7, line 221 Do the two paragraph  have the same title?

Response:

Thank you very much for your comment. The manuscript has been corrected.

page 6, line 198 Better: "The other 27 strokes occurred at the time of SLE diagnosis or later"

Response:

Thank you very much for the remark. We have revised the sentence as recommended and highlighted the change in the manuscript using a different color.

page 6, line 199: ...after the SLE diagnosis... Delete "of"

Response:

Thank you very much for your comment. The manuscript has been improved, accordingly.

page 11, line 282: ".. demonstrate insights of the usage .."    ??????

Response:

Thank you very much for your suggestion. The sentence has been improved.

page 11, lines 297-298: again I repeat that hypercholesterolemia is not a feature of CHA2DS2-VASc score

Response:

Thank you very much for your comment. We have removed hypercholesterolemia and replaced it with the factors evaluated in the CHA2DS2-VA score.

page 11, line 301 : internal comorbidities ?????

Response:

Thank you very much for your suggestion. We have removed the word "internal" to avoid any misunderstanding.

page 11, lines 307-308: please specify that HR is for stroke

Response:

Thank you very much for your comment. We have specified that this hazard ratio was related to ischemic stroke.

page 12, line 311: hazard for stroke?

Response:

Thank you very much for your comment. The hazard ratio (HR) refers to the earlier stroke onset. We have clarified that issue in the improved version of the manuscript.

page 12, lines 313-314 : sentence to be rephrased

Response:

Thank you very much for your remark. The sentence has been rephrased.

page 12, lines 322-327: all this part should be clarified

Response:

Thank you very much for your remark. The paragraph has been clarified.

page 12, lines 338-339: obviously I do not agree

Response:

Thank you very much for your comment. We have revised the conclusions by rephrasing the sentence with a novel observation in the context of SLE and the CHA2DS2-VA score.

We hope that the current version of the manuscript is suitable for publication. Once again, thank you very much for your valuable comments and suggestions.

Round 2

Reviewer 2 Report

Comments and Suggestions for Authors

Thank you for your effort in reviewing the text. The paper is much improved.

I have a last concern:

you consider together the group of patients with score 0-3 but in reality no stroke is reported in patients with score 0, 1 and 2. All the difference is due to patients with score 3 vs those with a higher score.

In my opinion you should report 1- that a different behavior was shown in patients 0-2 vs those >/= 3 as expected but 2- that analysing this group  those with score 3 had a higher rate of early onset of stroke

Author Response

Manuscript ID: jcm-3542434

Title: “Thromboembolic episodes in patients with systemic lupus erythematosus without atrial fibrillation/atrial flutter are related to the presence of at least 3 points in the CHA2DS2-VA score: a comprehensive retrospective analysis of 787 patients”

Reviewer 2

Thank you for your effort in reviewing the text. The paper is much improved.

I have a last concern:

you consider together the group of patients with score 0-3 but in reality no stroke is reported in patients with score 0, 1 and 2. All the difference is due to patients with score 3 vs those with a higher score.

In my opinion you should report 1- that a different behavior was shown in patients 0-2 vs those >/= 3 as expected but 2- that analysing this group  those with score 3 had a higher rate of early onset of stroke

Response:

Thank you very much for your valuable and constructive comment.

We fully agree with your remark and have revised the manuscript accordingly. Specifically, we now clearly distinguish between patients with scores of 0–2, score 3, and > 3 in the text, tables, and figures.

We have updated the relevant table to present the clinical characteristics of patients stratified into the following subgroups: 0–2 points, exactly 3 points, and > 3 points. This allows a more detailed analysis of the group in which no events were observed, as well as the group where such events began to occur. Indeed, we highlighted that our analysis revealed a higher rate f early ischemic stroke onset in those non-AF/AFL SLE patients with 3 points as compared to patients with more than 3 points in the CHA2DS2-VA score.

Additionally, we have revised the title of the manuscript to emphasize this novel and important finding.

We hope that the revised version of the manuscript addresses your comment is now suitable for publication. Once again, thank you for your insightful feedback, which helped to improve the clarity and accuracy of our paper.